# Immigration rates and species niche characteristics affect the relationship between species richness and habitat heterogeneity in modeled meta-communities

Avi Bar-Massada

Department of Biology and Environment, University of Haifa, Kiryat Tivon, Israel

Corresponding author
Avi Bar-Massada,
barmassada@gmail.com

## ABSTRACT

The positive relationship between habitat heterogeneity and species richness is a cornerstone of ecology. Recently, it was suggested that this relationship should be unimodal rather than linear due to a tradeoff between environmental heterogeneity and population sizes. Increased environmental heterogeneity will decrease effective habitat sizes, which in turn will increase the rate of local species extinctions. The occurrence of the unimodal richness–heterogeneity relationship at the habitat scale was confirmed in both empirical and theoretical studies. However, it is unclear whether it can occur at broader spatial scales, for meta-communities in diverse and patchy landscapes. Here, I used a spatially explicit meta-community model to quantify the roles of two species-level characteristics, niche width and immigration rates, on the type of the richness–heterogeneity relationship at the landscape scale. I found that both positive and unimodal richness–heterogeneity relationships can occur in meta-communities in patchy landscapes. The type of the relationship was affected by the interactions between inter-patch immigration rates and species' niche widths. Unimodal relationships were prominent in meta-communities comprising species with wide niches but low inter-patch immigration rates. In contrast, meta-communities consisting of species with narrow niches and high immigration rates exhibited positive relationships. Meta-communities comprising generalist species are therefore likely to exhibit unimodal richness-heterogeneity relationships as long as low immigration rates prevent rescue effects and patches are small. The richness-heterogeneity relationship at the landscape scale is dictated by species' niche widths and inter-patch immigration rates. These immigration rates, in turn, depend on the interaction between species dispersal capabilities and habitat connectivity, highlighting the roles of both species traits and landscape structure in generating the richness–heterogeneity relationship at the landscape scale.

## INTRODUCTION

One of the fundamental ecological concepts is that heterogeneous habitats can support more species, thus there is a positive relationship between species richness and habitat heterogeneity (*MacArthur & MacArthur, 1961*; *Cody, 1981*). However, a recent study has challenged the ubiquity of the positive richness–heterogeneity relationship, and suggested that the relationship should in fact be unimodal (*Allouche et al., 2012*). The reasoning behind this theory, termed the area–heterogeneity tradeoff, is that as habitats become increasingly heterogeneous in finite geographical space, the area comprising a given set of environmental conditions becomes smaller. Consequently, population sizes decrease, eventually leading to an increased prevalence of stochastic extinction events, and subsequently a lower overall species richness (*Kadmon & Allouche, 2007*). The area-heterogeneity tradeoff, therefore, has three main predictions: (1) there is a negative relationship between population sizes and habitat heterogeneity; (2) there is a positive relationship between habitat heterogeneity and local extinction rates; and therefore (3) there is a unimodal relationship between species richness and habitat heterogeneity. However, in different systems it is possible to find positive, unimodal, or even negative relationships between richness and heterogeneity, depending on the characteristics of the species in those systems, such as their niche widths (*Allouche et al., 2012*), their fecundity (*Kadmon & Allouche, 2007*), and the rate of immigration into the local community from the regional species pool (*Kadmon & Allouche, 2007*); as well as the hierarchical scale of the analysis (*Bar-Massada & Wood, 2014*), the size of the local habitat (*Kadmon & Allouche, 2007*), and the environmental variable whose heterogeneity is measured (*Bar-Massada & Wood, 2014*).

The original area-heterogeneity tradeoff hypothesis (*Kadmon & Allouche, 2007*) was developed for communities inhabiting a single island (or habitat patch) in an island-mainland system, based on the unification of niche theory (*Hutchinson, 1957*) and the theory of island biogeography (*MacArthur & Wilson, 1967*). The first empirical test of the theory (*Allouche et al., 2012*) was based on analyses of breeding bird species in geographical units of 100 km$^2$ in Spain, where elevation range served as a measure of habitat heterogeneity. This study, therefore, did not account for the spatial structure (e.g., patchiness, connectivity) of the geographical regions which served as sampling units. Moreover, elevation range, although a popular measure of habitat heterogeneity (*Veech & Crist, 2007*; *Allouche et al., 2012*), is a simplistic and indirect measure of actual habitat heterogeneity (*Hortal et al., 2013*), as species respond and utilize habitat features at much smaller spatial scales (*Bar-Massada & Wood, 2014*).

It is possible, however, to adapt the area-heterogeneity tradeoff theory to a broader-scale, spatially explicit framework, which is more in line with the traditional landscape ecological view of landscapes as comprising patches of different types located within a non-habitat matrix. Consider a landscape consisting of multiple patches of different types. The classic prediction would be that as landscape heterogeneity increases (i.e., there is an increase in patch richness or the number of patch types, and an increase in patch evenness, which corresponds with an increasingly uniform areal distribution of different patch types), species richness increases as well. However, as landscape heterogeneity keeps

increasing (by the addition of more and more patches and patch types), the patch size distribution shifts to the left and many patches become smaller, while at the same time edge density increases (*Fahrig et al., 2011*). Consequently, their ability to support sufficiently large populations of individual species diminishes. Furthermore, with increased landscape heterogeneity, smaller patches may become more isolated, and consequently the landscape becomes more fragmented, with potential detrimental effects on species left in patches. Thus, the intermediate heterogeneity hypothesis (*Fahrig et al., 2011*) suggests that at extremely high heterogeneity levels, the detrimental effects of fragmentation offset the positive effects of landscape heterogeneity, resulting in decreasing species richness at the landscape scale. Yet species in patchy landscapes often belong to meta-populations, and local communities in patches are part of a meta-community (*Leibold et al., 2004*). In both meta-populations and meta-communities, species persistence at the landscape scale can be maintained by source–sink dynamics and rescue effects (in which populations in small sink patches or patches with suboptimal habitat conditions are maintained by immigration from neighboring source patches; *Brown & Kodric-Brown, 1977*; and mass effects (*Shmida & Wilson, 1985*; *Kunin, 1998*), through which species can persist in suboptimal habitats by dominating the propagule pool. Indeed, *Kadmon & Allouche (2007)* showed that higher immigration rates from mainland to island communities promote positive richness-heterogeneity relationships within single communities. In a meta-community/patchy landscape context, the question becomes: do higher immigration rates among patches (regardless of the presence of a mainland) maintain the persistence of species in meta-communities, thus promoting positive richness-heterogeneity relationships across different landscapes? Moreover, species niche widths can affect the number of species that can establish in patches, as species with wider niches can establish in more patch types; if more species establish in a patch, there is less area available for each species (*Tilman, 2004*). It is possible therefore that niche width can interact with inter-patch immigration rates to affect overall species richness, as well as the type of the richness heterogeneity relationship.

Here, I developed a spatially explicit meta-community model, based on existing modeling approaches in community ecology (*Gravel et al., 2006*; *Bar-Massada, Kent & Carmel, 2014*; these models were developed to simulate the roles of dispersal and demographic stochasticity in driving species assembly in single communities), to test whether the area-heterogeneity tradeoff can generate a unimodal relationship between species richness and habitat heterogeneity for meta-communities across different landscapes. The model simulates the dynamics of a meta-community residing in a patchy landscape that is characterized by a given level of environmental heterogeneity. Model results reflect the joint operation of three mechanisms that generate species richness gradients along heterogeneity gradients: (1) niche filtering, by assigning patches different environmental conditions, making them suitable for only a subset of species in the meta-community; (2) interspecific competition for space, according to species niche requirements and their relative abundances in the propagule pool; and (3) area-heterogeneity tradeoffs. As patch sizes are finite, any increase in landscape heterogeneity results on a decrease in the

area available for each species, leading to smaller population sizes, and an increase in the likelihood of stochastic extinctions.

Specifically, I hypothesized that: (1) meta-communities comprising species with higher immigration rates are likely to produce the classical, positive richness–heterogeneity relationship, as species are less likely to go extinct at the landscape scale; (2) low immigration rates preclude rescue effects, making local population extinctions irreversible, and promoting a unimodal richness-heterogeneity relationship at the landscape scale; (3) finally, increasing species' niche width would promote more unimodal relationships, as population sizes in finite-area patches will become smaller, increasing the rate of stochastic extinctions within patches.

## METHODS

### The model

The model simulates the spatiotemporal dynamics of species in a meta-community located on a patchy landscape. The landscape consists of $J$ sites distributed among $k$ patches, with the environment $E$ in each patch being unique. $E$ is a one dimensional variable, which ranges from 1 to 500. Prior to each model simulation, patches are assigned random locations across a two-dimensional landscape spanning 100-by-100 arbitrary distance units. Each patch consists of $J_p$ sites having the same environment $E$, and each site hosts a single individual. The value of $J_p$ in each patch is drawn at random from a multinomial distribution with equal probabilities for all patches, while ensuring that the sum of $J_p$ values across the entire landscape equals exactly to $J$. Consequently, the number of sites can vary slightly across patches.

At the first time step of a model simulation, a meta-community of $S$ species and $J$ individuals is allocated across the landscape, with each patch having $J_p$ individuals (possibly of multiple species) selected at random from the entire species pool according to the suitability of species to environmental conditions in patches. All species are demographically equivalent (have identical birth and death rates), and birth rates are very high, so that each site receives an influx of propagules. At any subsequent time-step, a proportion $d$ of individuals die and their sites become available for colonization by new individuals. New individuals comprise ones coming from other sites in the same patch, as well as immigrants from other patches in the landscape. Individuals arriving in a patch compete for establishment in available sites according to a lottery process (*Chesson & Warner, 1981*). The number of establishing individuals from different species, $N'_{i,t}$, is drawn from a multinomial distribution with a probability $R_i$ for each species, with a sample size that equals the total number of individuals that died in the patch. $R_i$ depends on a species' relative abundance in the propagule pool coupled with the fit of the environment in the site to its niche requirement:

$$R_i = \left[ \frac{f_i N^*_{i,t}}{\sum_j f_j N^*_{j,t}} \right] \tag{1}$$

$N_{i,t}^*$ is the total number of propagules of species $i$ competing for establishment in available sites (including immigrants from the meta-community), and $f_i$ is a measure of fitness, denoting the suitability of environmental conditions in the patch to the niche requirement of species $i$ (*Tilman, 2004*; *Gravel et al., 2006*; *Bar-Massada, Kent & Carmel, 2014*):

$$f_i = e^{-\left[\frac{(e_i - E)^2}{2\sigma^2}\right]}$$ (2)

where $e_i$ is then niche optimum of species $i$, $E$ the environmental conditions in the patch, and $\sigma$ a measure of niche width. Once the number of establishing individuals from each species is quantified using draws from the multinomial distribution, the abundance of species $i$ in a patch following mortality and establishment is:

$$N_{i,t+1} = \lfloor (1 - d_i) N_{i,t} \rfloor + N_{i,t}'$$ (3)

where $N_{i,t}$ and $N_{i,t+1}$ are whole numbers (integers) that denote the total abundance of species $i$ in the patch in the current and previous time step, $N_{i,t}'$ is the number of establishing individuals following dispersal events, $d_i$ is mortality rate of species $i$, and the brackets around the first term in the right hand side of the equation represent a floor function that ensures that species abundance is an integer.

The abundance of propagules of species $i$ competing for establishment in a given patch ($N_{i,t}^*$ in (1)) is calculated by multiplying the vector of species $i$ abundance in all patches by the probability of propagule arrival in the patch (from both local and neighboring patches).

$$N_{i,t}^* = \sum_{j=1}^{k} p_j N_{i,j,t}$$ (4)

where $p_j$ denotes the probability of propagule arrival in the focal patch from patch $j$, $N_{i,j,t}$ is the abundance of species $i$ in patch $j$ at time $t$. Assuming the same dispersal capabilities for all species, in a landscape with $k$ patches the probability of propagules arriving into patch $i$ from patch $j$ ($p_{i,j}$) depends on the distance $d_{ij}$ between them, and is quantified using a negative-exponential dispersal kernel:

$$p_{i,j} = e^{-z d_{ij}}$$ (5)

where $z$ is the rate coefficient, which denotes the rate of decrease in arrival of propagules with increasing distance (larger values correspond with decreased arrival of propagules from distant patches). In this setting, the value of $z$ serves as a measure of inter-patch immigration rates, with lower values corresponding with increased immigration rates from other patches in the meta-community.

## Model simulations

I developed, tested, and analyzed the model and its results in R (*R Core Team, 2013*; R scripts appear in the Supplemental Information 4). A flowchart depicting the modelling

process appears in Fig. S1. In each model run, I generated a landscape that consisted of 500 sites, grouped into a randomly assigned number of patches between 2 and 500, with each patch having a unique environmental condition $E$ between 0 and 500. All patches had roughly the same size (barring minor rounding effects), and therefore the diversity of environmental conditions increased together with the number of patches, while the fractional cover of each environmental condition was close to constant when the number of patches was very large. This yielded a complete range of compositional heterogeneity levels (from two large patches with two $E$ values to 500 small patches with 500 different $E$ values, denoting highly homogeneous and highly heterogeneous landscapes, respectively). Given that the fractional cover of each environmental condition was close to (but not completely) constant at maximum heterogeneity (cover type evenness, sensu *Fahrig et al., 2011*), I used Shannon's index of landscape diversity (*Nagendra, 2002*) as the measure of compositional environmental heterogeneity (but see *Biswas & Wagner, 2012* for a discussion on measures of landscape heterogeneity in the meta-community context). Shannon's index of landscape diversity is denoted by:

$$\text{Shannon's diversity} = -\sum_E \frac{J_E}{J} \ln\left(\frac{J_E}{J}\right) \tag{6}$$

where $J_E$ is the number of sites in all patches of a given type ($E$ value). In addition to Shannon's index, I also quantified heterogeneity using patch richness (the number of unique $E$ values).

At the beginning of each simulation, the meta-community consisted of 250 species. Each species was randomly assigned a niche optimum value ($e$) from the list of $E$ conditions that already existed in patches (to prevent species from having a niche optimum that does not fit conditions in available patches). Species had the same niche width parameter ($\sigma$). Death rate $d$ was 0.25 and identical for all species. In all cases, models were run for 1,000 time steps, after ensuring that changes in overall species richness were negligible at this stage by comparing them to the results of identical analyses with 250 time steps (Figs. S2 and S3). Landscape scale species richness was calculated at the end of each run.

During model simulations, I pre-determined niche width and immigration rates ($z$ in Eq. (5)) to assess their effects on the relationship between meta-community species richness and landscape scale environmental heterogeneity. I tested all possible combinations of $\sigma = 1, 5, 10$, and 50 (from very narrow to wide niches, respectively). To alter immigration rates, I ran the simulations with four different values of $z$: 0.2, 0.1, 0.05, and 0.025, which correspond with increasing rates of inter-patch immigration, respectively (Fig. 1).

## RESULTS

The type of the richness-heterogeneity relationship at the landscape scale was affected by complex interactions between niche width and inter-patch immigration rates. In general, I found two types of relationships, positive and unimodal. When species had narrow niches, the relationship between richness and heterogeneity was positive regardless of inter-patch immigration rates, and this result was consistent for both heterogeneity metrics

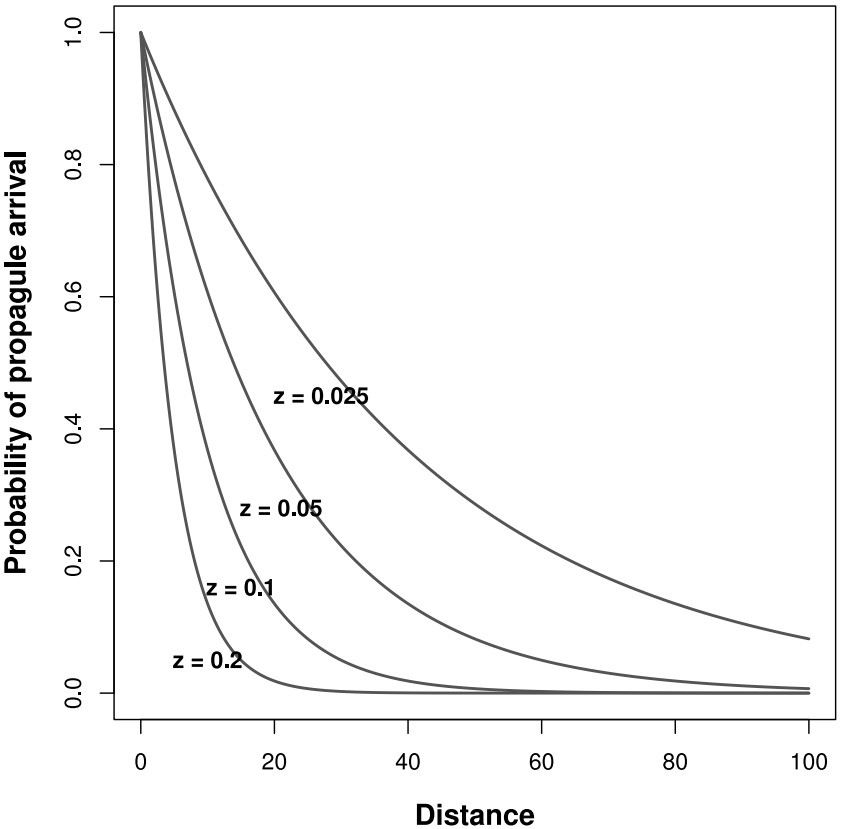

**Figure 1** **Effect of inter-patch distance on the probability of propagule arrival into a patch.** Each curve is based on a different *z* parameter (Eq. (5)).

(Figs. 2A and 2E). In general, meta-communities with higher immigration rates had lower species richness levels overall (Fig. 2). Moreover, the difference in overall species richness among meta-communities with different immigration rates increased with landscape heterogeneity (Figs. 2A and 2E). When heterogeneity was low, richness was very low and all meta-communities had similar species richness levels. In contrast, when heterogeneity was maximal (500 patches, each one having a unique environment *E*) species richness in meta-communities with low inter-patch immigration rates had at least twice as many species compared to meta-communities with high inter-patch immigration rates (Fig. 2A).

As species niche width increased, the dominance of the positive richness–heterogeneity relationship started to diminish, and unimodal richness-heterogeneity relationships emerged in meta-communities with low to intermediate inter-patch immigration rates (Figs. 2B–2D and 2F–2H). Meta-communities with high immigration rates retained the positive relationship type, but their overall species richness was very low compared to all other meta-communities. In general, meta-communities with low immigration rates had higher levels of species richness compared to those with high immigration rates. To conclude, unimodal richness–heterogeneity relationships at the landscape scale were prominent in meta-communities comprising species with wider niches (more generalists)

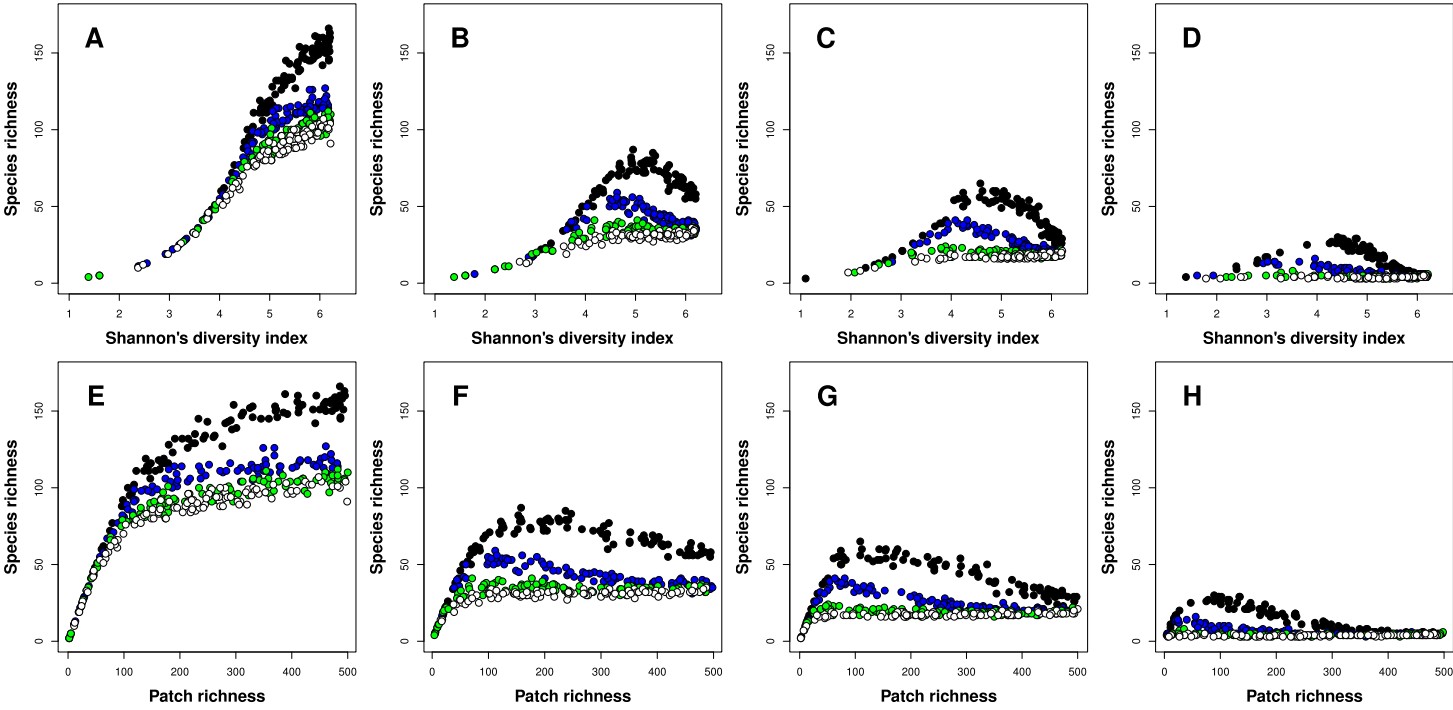

**Figure 2 Relationships between species richness and landscape heterogeneity for modeled meta-communities.** (A–H) correspond with different species niche widths (A, E—very narrow, B, F—narrow, C, G—intermediate, and D, H—wide). Curves denote inter-patch immigration rates, with circle colors depicting the value of the z parameter (0.2—black, 0.1—blue, 0.05—green, and 0.025—white, reflecting increasing levels of inter-patch immigration rates). The top row is based on Shannon's measure of heterogeneity, while the bottom row uses patch richness as the heterogeneity measure.

but low to intermediate inter-patch immigration rates (or more fragmented landscapes). These results were consistent between both heterogeneity metrics, but patch richness exhibited right-skewed unimodal relationships, compared to the left-skewed unimodal relationships which emerged when Shannon's diversity was the heterogeneity measure.

## DISCUSSION

The results of this analysis highlight the potential coupled roles of species niche characteristics and dispersal capabilities in dictating the type of the richness–heterogeneity relationship for meta-communities. Unimodal relationships emerge when locally extinct populations cannot re-emerge due to the lack of rescue effects from neighboring patches. This results in the eventual extinction of species from the entire meta-community, leading to a decrease in species richness. Species' niche width affects the type of the relationship via controlling the number of species that can establish in a patch. In heterogeneous landscapes where patches differ in environmental conditions, generalist species are able to establish in more patches compared to specialists. Consequently, patch-scale species richness is expected to increase and the area available for populations within patches becomes smaller, increasing the likelihood of stochastic extinction. In patches of finite areas, therefore, mechanisms that initially contribute to increased species richness

(i.e., wide niches) are the very same drivers of local extinction, as long as patch isolation prevents rescue effects and patches are relatively small.

The effect of niche width on the shape of the richness-heterogeneity relationship are in line with the empirical findings of *Carnicer et al. (2013)*, who re-analyzed the data in *Allouche et al. (2012)*. They reported that assemblages of species with narrow niches exhibited positive linear relationships with habitat heterogeneity, as I found in this study (Fig. 2A). *Carnicer et al. (2008)* and *Carnicer et al. (2013)* suggest that for species with narrow niches, the traditional niche filtering process is what drives species richness gradients, leading to positive richness-heterogeneity relationships. In line with my other theoretical findings (Figs. 2B–2D), *Carnicer et al. (2013)* found that unimodal richness-heterogeneity relationships emerge when species have wide niches. Notice, however, that *Allouche et al. (2013)* criticized the approach of *Carnicer et al. (2013)* for not using analogous measures of heterogeneity and niche breadth (i.e., they used habitat preference as a measure of niche width, and elevation range as a measure of heterogeneity). My analysis is robust to this criticism as both niche breadth and environmental heterogeneity were quantified based on the same environmental variable.

The finding that species richness is lower when immigration rates are high is in line with the suggestion of *Fahrig et al. (2011)* (based on *Amarasekare, 2008*) that reduced inter-patch immigration rates lead to de-coupling of patch dynamics, resulting in meta-community persistence via increased species coexistence. Here, patch-scale species richness is the outcome of the interaction between relative fitness differences among species (*Chesson, 2000*; *Adler, HilleRisLambers & Levine, 2007*), which promote the dominance of the best-adapted species; and mass effects (*Shmida & Wilson, 1985*; *Kunin, 1998*), which allow less-suitable species to persist since they are abundant in the propagule pool. Presumably, there are two opposite scenarios that describe the effect of immigration rates on this interaction. First, when immigration rates are low and a patch consists of a large population of less-suitable species (versus a small population of a better-suited species), they are able to persist via within-patch mass effects as long as propagules of the better-suited species cannot arrive in the patch. In contrast, when immigration rates are high, a small population of a less-suitable species can persist in a patch even as it is dominated by a superior species as long as a sufficient flux of its propagules keeps arriving in the patch, promoting a rescue effect (*Brown & Kodric-Brown, 1977*; *Leibold et al., 2004*). However, in my model all species had the same dispersal rates and demographic parameters, thus the overall effect of unlimited immigration between patches is the arrival of the competitively-optimal species to every suitable patch. In the absence of temporal variation in patch condition or species demographical traits (i.e., birth and death rates), the eventual outcome of any lottery-type competition between species is the dominance of the most-suitable species, with all other species driven to local extinction (*Chesson, 2000*; *Gravel, Guichard & Hochberg, 2011*). Therefore, when there is no inter-patch immigration overall species richness at the meta-community scale will converge towards the number of patches types (i.e., patch richness), as long as for each patch type there is a species whose niche requirements fit the conditions in the patch better than all other species.

I caution that model results should be interpreted in the context of the scale of the analysis, coupled with richness of the regional species pool. The outcome of the area-heterogeneity tradeoff is more likely to be detected in empirical studies when species are sampled in small (and area-limited) habitat patches. As habitat patches become larger, so does the number of potential species they can host without portraying the limiting factor of habitat area. Thus the type of the richness-heterogeneity relationship is likely to be affected by the size of the regional species pool compared to the area available in local patches and landscapes. The relationship will also be affected by the ability of populations to persist in small patches. Communities comprising species that cannot persist in small patches are likely to exhibit the unimodal relationship, while communities comprising species that can are likely to portray positive richness-heterogeneity relationships.

In general, the results of this model (and the empirical findings of *Bar-Massada & Wood, 2014*) raise an intriguing question about the length of the heterogeneity gradient in actual landscapes: to reach truly low levels of species richness that are caused by increased extinctions at very high heterogeneity levels, how heterogeneous should a real landscape be? Obviously, the conditions used in this study to represent the extreme end of the heterogeneity gradient (500 patches of different types, each comprising a single site) are unrealistic. Yet the richness-heterogeneity relationship became negative at much lower levels of heterogeneity than the theoretical maximum. Understanding its drivers and predicting the tipping point of the richness-heterogeneity relationship in actual landscapes is an intriguing question for future studies, which may offer useful insights for conservation planning.

## CONCLUSIONS

The ongoing debate about the predominant type of the richness–heterogeneity relationship may be overcome if we better understand the processes that drive this relationship for different taxa, habitats, and spatial scales. This study, although based solely on a theoretical model, suggests that at broad spatial scales the type of this relationship may be driven by an interaction between species niche characteristics and their dispersal capabilities. Ultimately, if we were able to better understand the mechanisms that drive the richness-heterogeneity relationship, including those caused by human activity (*Seiferling, Proulx & Wirth, 2014*), it would add invaluable insight for conservation management and planning, by informing about the level of heterogeneity that may support maximum levels of biodiversity in a given landscape.

## ACKNOWLEDGEMENTS

I thank Cajo Ter Braak, Hans Baveco, Laura Graham, and an anonymous reviewer for their insightful comments and suggestions that greatly improved this manuscript.

### Funding

This project received no external funding.

## Competing Interests

The author declares there are no competing interests.

## Author Contributions

- Avi Bar-Massada conceived and designed the experiments, performed the experiments, analyzed the data, contributed reagents/materials/analysis tools, wrote the paper, prepared figures and/or tables, reviewed drafts of the paper.

## Supplemental Information

Supplemental information for this article can be found online at http://dx.doi.org/10.7717/peerj.832#supplemental-information.

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
