# Peer review of "Immigration rates and species niche characteristics affect the relationship between species richness and habitat heterogeneity in modeled meta-communities"

_PeerJ, doi:10.7717/peerj.832_

## Round 0.1 · original submission · Major Revisions

General

From my own reading, I found that, in agreement with reviewer 1, some ad-hoc aspects in the simulation setup are not very clearly written. Also, the way niche width and E are defined (line 134 and line 151) make niche width related to environmental range, so confusing/confounding width and landscape heterogeneity. Also, the point for reviewer 1 on demographic stochasticity is important and I would like to add the question whether it matters whether N_it is a real, truncated to integer or taken as a Poisson random variable.
Shannon is a measure of landscape heterogeneity for unordered patch types. Your environment is quantitative and one-dimensional (values between 0 and 500). So an alternative measure is (related to) variance and, if the model would be spatially explicit, spatial autocorrelation. Please comment.

The simulation program should made public.

Details

1. In abstract: after niche characteristics are specialized to niche width, use the word niche width instead of niche characteristics. I did not understand the last sentence after “which”. Rephase or delete.
2. lines 17-20 Unimodal relationships naturally follow from combination of an increasing and an decreasing relationship [see enzyme activity-temperature relationships]. So reorder the points here with [1] reported last.
3. line 17. Are there predictions or hypotheses?
4. line 66-67. Summarise theory with references for niche width and number of species and describe the direction of the relationship.
5. line 88. “Type” and s are not mentioned later on. Why here???
6. line 89. Say here that E is one-dimensional (as needed for equation [2] and values 0-500).
7. line 89. J_p (p=1,..,n). How was J_p determined?
8. line 102. The formula gives all, not just “the number of establishing individuals from different species is determined according to their relative abundance in the propagule pool coupled with the fit of the environment in the site to their niche requirements”. So close with a period. And restart for example with: Let N….., then …
9. line 103/109. Where -> where as an equation is just part of a sentence.
9A. Anything taken random here??
10. S is species richness? With a fixed array of species indexed by i, S cannot be the number of species at the patch [as for j = S may need to indicate different species in patches with the same S]. Sum over all species, which is no problem as N_jt = 0 for all species absent in the patch.
11. It is the left-hand term of the right-hand side of equation [1]. Rephrase.
12. line 101-113. Reconsider a rearrangement.
13. line 117 k patches?? It were n on line 88.
14. line 118, equation [3]. Subscript l is undefined.
15. Eq. [3]. The sum over all p is one, making it probabilities, that mimic a dispersion kernel centred on a patch. But on line 147 you use a different scaling. ???
16. Line 114. Give an explicit formula later on or in supplementary material. Anything taken random here??
17. line 132. I would strongly advice that the program is supplied as supplementary material. This allows verification and speeds up science!
18. line 136-137. Do I understand correctly that patch size is constant across the simulated landscape [and varies among landscapes].
19. line 142. If Shannon = -sum(p*ln(p), then with p = 1/n, n = 2:500, I expect max Shannon = 25 whereas the figures give a range of 1-6. If I change to log10, then indeed, the range is 0.15-8.35. So what do you specify?
20. line 140-142. Given… I used... Rephrase. After Given, I expect that you say that the evenness is constant and only the number of different E values matter, so any reasonable measure of diversity would hold and you chose Shannon. You could have used just the number of E values!
21. “according to E conditions that already existed in patches (to prevent species from having a niche optimum that does not fit conditions in available patches).” How is that implemented and how does that change the meaning of niche width?
22. line 158. The values 1-10 for sigma are very small considering the E values of 0 – 500. With the full 500 range, even sigma = 10 gives a standard deviation of species turnover of 500/10 = 50 sd unit (http://library.wur.nl/WebQuery/wurpubs/436947 )which is very unrealistic. A value of 10 is about maximum ever observed I would think.
23. line 161-162 How are the values 0,0.5 1, 10,100 and 1000 used to fill/generate P?

·

Basic reporting

No comments

Experimental design

Regarding the applied method: some information that would make the research reproducible I find missing from the text, and should be included
- It is not always clear which values were used (see detailed comments)
- It is also not always stated whether variables were treated as deterministic or stochastic (see detailed comments). This is especially relevant for understanding the risk of local extinction. A related issue is whether whole individuals were considered or not. I assume this was the case (otherwise demographic stochasticity would be hard to represent) but it should be explained in the text.
- Related to local extinction: does the model allow empty patches to arise (as in multi-species metapopulation models)? If that is the case the fraction of empty patches should also be shown in the results. Or less extreme, does it allow partly filled patches, vacancies that can’t be filled because the absolute number of propagules (from inter- and intra-patch immigration) is too small? Should depend on the birth rate, but birth rate is not given.
- In general, how did the author deal with variation? How many landscapes were considered, how many replicate simulations per landscape were tested, do the dots in figure 1 represent averages or single simulation results, etc?

Validity of the findings

Two essential aspects need to be addressed more elaborately.

I am not sure temporal dimension can be ignored completely. All simulations were run for 250 time steps, and although it is stated (line 154) that it was ensured that changes in overall species richness were negligible after that period, I am not convinced that this really can be the case for all combinations of landscapes and immigration and niche width values. Are the observed values really equilibrium values? And how fast will this equilibrium be reached in the different landscapes/settings? Could it be that the uni-modal relationship arises because for some settings we have slower (within patch) dynamics (in terms of the speed with which species become dominant) than for others? (a trade-off between patch size (larger patches slower dynamics) and niche width+immigration (speeding up dynamics)? The author should include a prove that we are not – in some of the cases – looking at transient behaviour instead of equilibrium state. This is easily tested by increasing run time to 500 or 1000 steps and providing the results as in figure 1.

The way the final probabilities of propagule arrival were defined is not very clear. Is the matrix P really normalized always, how, and does it influence the results? And can the approach be considered spatially-explicit when random connectivity matrices are used? Why not calculate P from the generated landscapes? Can the landscape definition give rise to landscape-level structures (e.g., clusters of highly connected patches). A precise description and motivation of what was exactly done wrt P in all cases should help, plus a discussion of the potential consequences of the choices made.

Additional comments

Some comments by line number:
Line 43. Patch richness and evenness should be defined
Line 60 – 62. Not clear how to understand this sentence..
Line 70. Insert “a” (before “spatially explicit”)
Line 70-71. Short characterisation of what these “existing modeling approaches in community” should be inserted
Line 74. “many meta-communities” is not clear. Many communities considered at the same time? “different landscape” referring to the whole landscape, or the landscape around a local patch?
Line 88-89. How were the different landscapes generated? Are they really spatially-explicit, with explicit x,y locations of patches? And if so, was this explicitness really used in later steps (e.g. defining connectivity)?
Line 95. No value of the birth rate is given anywhere. Could it be that is was set to 1, to ensure the presence of sufficient propagules always?
Equation 1. N integer number? Vacant sites integer number? Will all vacant sites always be filled? (only the proportion of different species in the propagule pool seems to play a role, not the absolute size).
Line 115-116. Shouldn’t this be multiplied by the birth rate as well? (birth rate = 1?)
Line 118. Propagule, not propoagule
Equation 3. What is the reason to normalize probabilities in this way? Why would the sum of all entries in the matrix be one? Doesn’t this lead to very low probabilities pij in case of many patches?
Line 127-129. How were the entries in P defined then? Based on the usual negative exponential relationship with distance? (further on it says: randomly set...)
Line 142. Shannon’s index, although probably widely known, should be shown here.
Line 145. I am not sure a random immigration matrix P can give a consistent representation of the connectivity of a landscape? Can this potentially affect the outcome or not?
Line 147. This is contradictory to equation 3. This won’t normalize the matrix?
Line 154. Has that really been tested for all combinations of settings?
Line 159-162. And again some sort of normalization after changing the diagonal values?

Figure 1 could become clearer by using different symbols (and/or colors) to indicate which of the immigration cases the dots represent (0, 1/2 , 1, 10, 100 or 1000 at the diagonal of P). In the current figure it looks like there are 5 different cases, but there should be 6?

·

Basic reporting

The fundamental concept of the positive richness–heterogeneity relationship has recently been challenged and it has been suggested that this relationship is in fact unimodal in some cases. It is argued that the unimodal relationship is potentially because of an area–heterogeneity trade-off. The author has used a theoretical meta-community model to examine the effect of niche width and immigration rates on the richness–heterogeneity relationship. The author found that meta-communities consisting of species with wider niches and low inter-patch immigration rates are more likely to display a unimodal relationship, whereas meta-communities made up of species with narrow niche widths and high inter-patch immigration rates are more likely to display a positive richness–heterogeneity relationship.

The article is an interesting use of simulation models to start unpicking the mechanisms driving the shape of the richness–heterogeneity relationship. It is on the whole well written and the background to the study is well researched, with the findings set in the context of the literature well. One area of the study which needs improvement is the evaluation of the shape of the richness–heterogeneity relationship. (Further explanation in Experimental Design section)

Experimental design

The assessment of the shape of the relationship could be improved and made more robust by making this a quantitative assessment. In some cases it is quite difficult to make the distinction visually – for example in panel A with low inter-patch immigration, the relationship is said to be positive. In panel B with no inter-patch immigration this relationship is stated to be unimodal. To me these look fairly similar. It would add weight to the conclusions if the shape of this relationship was tested, for example by fitting linear and quadratic regression models and comparing model fit.

Validity of the findings

I am unsure until the quantitative assessment of the shape of the relationship has been done. The conclusions do make logical sense if the visual assessment was accurate.

Additional comments

Because the methods in this study are fairly complex, it might help to have some kind of table/flow-chart/diagram explaining how they work, perhaps as supplementary material. Particularly for the landscape design. Please also consider sharing your code to aid reproducibility.

General comment: Need to be careful with hyphens / en-rules. Make sure to remove spaces when talking about richness–heterogeneity; species-level; trade-off; area–heterogeneity etc.

The second sentence of the abstract isn't clear. Perhaps change to something along the lines of "Recently, it was suggested that this relationship should be unimodal rather than linear due to a trade off between environmental heterogeneity and population sizes. Increased environmental heterogeneity will decrease effective patch sizes, which in turn will increase local species extinctions."

Line 69: Should either be phrased as a question or the question mark removed.

The hypotheses (lines 75-83) should be set out clearer (perhaps numbered). Also, it is not explicitly clear that the second hypothesis relates to the unimodal relationship. Perhaps change the last sentence (lines 80-81) to “At the landscape scale, this effect should result in the reduction of overall species richness, and therefore a unimodal richness–heterogeneity relationship.”

The model simulation methods do not quite add up. I think that the issue here is on line 135 – from what I have understood it is possible this should read “All sites had roughly the same size”. If all patches have the same size, it would not be possible to get the range of heterogeneity levels in the way they are discussed on lines 138-140. This paragraph needs revising to be clearer.

Line 143 – take the reference out of the brackets because it already is contained with the brackets starting “but see”.

Reviewer 3 ·

Basic reporting

No comments

Experimental design

No comments

Validity of the findings

No comments

Additional comments

Review of “Immigration rates and species niche characteristics affect the relationship between species richness and habitat
heterogeneity in modeled meta-communities”


This is an interesting and valuable theoretical contribution, examining and contrasting the relative importance of area-heterogeneity relationships, habitat diversity, niche-filtering processes and niche width effects in gradients of species richness and landscape heterogeneity. The manuscript is interesting but in the introduction it possibly needs a more precise definition of the different ecological mechanisms or main hypotheses included in the model. In other words, the model simulates several ecological mechanisms but these mechanisms are not clearly explained and defined in the introduction. For example, in the introduction the authors mainly focus on area-heterogeneity trade-offs but alternative hypotheses included in the model are not considered. Finally, I suggest that recent empirical tests of the area-heterogeneity trade-offs are in line with some of the results presented in Figure 1 and this could be discussed or briefly commented.

Below I provide a point-by-point list of the major points that could be addressed in a revised version of the manuscript.

1. In the introduction please provide in a table or a paragraph a precise list of the different mechanisms and hypotheses included in your model and simulations. It seems to me that the model is not only simulating area-heterogeneity trade-offs, but also includes alternative mechanisms associated with classical hypotheses of biodiversity maintenance:

Hypothesis 1: Niche filtering and habitat diversity hypothesis. The model generates gradients of habitat diversity, simulates niche-filtering processes and results in the emergence of species richness gradients. Of note, in some simulations this is possibly achieved without the necessary operation of area-heterogeneity trade-offs (for example Figure 1A). As stated in Line 167 by the authors ”When species had narrow niches, the relationship between richness and heterogeneity was positive regardless of inter-patch immigration rates”. So it seems that linear species richness gradients associated with increased habitat heterogeneity emerge without the necessary operation of area-heterogeneity trade offs (increased population extinction rates due to reduced habitat area in high heterogeneity sites).

Hypothesis 2: Competition processes and niche width effects. Consistent with classical niche theory, increased niche width reduce overall richness and coexistence in the system (compare Figures 1A, B, C). In addition, it seems to me that low or zero inter-patch migration rates implicitly confer or simulate higher competitive ability to local patch populations (because their intra-patch local demographic rates or recruitment probabilities are favored respect to the demographic dynamics of potential invasive species in the metacommunity, increasing the probability of local persistence and reducing extinction rates). Therefore, species richness trends are affected by niche width effects and by inter-patch migration rates (a parameter that could be interpreted as a surrogate for dispersal, local competence, and invasion capacity). Critically, as with hypothesis 1, these variables (niche width and inter-patch dispersal rates) can substantially alter observed species richness trends without the necessary operation of area-heterogeneity trade-offs.

Hypothesis 3: Area-heterogeneity trade-offs. Reduced habitat area with increased heterogeneity is expected to increase extinction rates at high heterogeneity sites and produce hump-shaped patterns. Critically, in your model model area effects are mixed (or interacting with) with niche width settings and invasion/competition effects (intra-patch dispersal parameter). So, several ecological hypotheses are examined in the simulations (H1-H3). Interestingly, the simulations suggest that robust hump-shaped trends only emerge for generalist species (wide or intermediate niche) at high or inter-mediate dispersal rates.

It would be a great improvement if you could clarify, separate or quantify the relative importance of these hypotheses and factors (i.e. niche width, inter-patch dispersal and habitat area effects) on the observed species richness trends.

2. In the discussion it would be certainly interesting to contrast your theoretical results with the observed trends in recent tests of area-heterogeneity trade-offs. For example, in your model, niche-filtering processes and habitat diversity generate a linear increase of species richness with increased habitat heterogeneity (Fig 1A, species characterized by narrow niches). This pattern seems to some extent independent of the operation of area-heterogeneity effects, and robust to inter-patch migration effects. In line with this result, in a recent empirical test of area-heterogeneity trade-offs, a linear relationship in narrow niche specialists has been reported (see Figure 2 E, F in Carnicer et al 2013). These trends have been specifically linked to the niche-filtering hypothesis (please see Carnicer et al 2008, 2013). Overall, your theoretical results (Fig 1A) and the reported empirical trends (Carnicer et al 2013, 2008) suggest that in narrow niche species niche filtering processes may often predominate in shaping species richness gradients.
Similarly, a recent test of area-heterogeneity trade-offs reported hump-shaped trends only in wide-niche species, also in line with your theoretical simulations (Figure 2 A,B,C,D in Carnicer et al 2013). This could be discussed. Overall, all these results suggest that area-heterogeneity trade-offs are not the main mechanism underlying linear species-richness/heterogeneity trends in narrow niche species (habitat specialists) and points to niche-filtering processes and habitat diversity effects (Carnicer et al 2008, 2013, and see Carnicer et al 2012 for a definition of niche-filtering).

Finally, a recent review by Stein et al in Ecology Letters could be also discussed, cited or commented (Environmental heterogeneity as a universal driver of species richness across taxa, biomes and spatial scales, Eco Let.)



Minor comments:


Line 183. “To conclude, unimodal richness – heterogeneity relationships at the landscape scale were prominent in meta-communities comprising species with wider niches (more generalists) but low inter-patch immigration rates (or more fragmented landscapes).” It seems to me that in your figures hump-shaped trends emerge at intermediate inter-patch immigration rates. If considered appropriate, replace “low” by “intermediate”.

Line 193. “Species’ niche width affects the type of the relationship via controlling the number of species that can establish in a patch. In heterogeneous landscapes where patches differ in environmental conditions, generalist species are able to establish in more patches compared to specialists. Consequently, patch-scale species richness is expected to increase and the area available for populations in patches becomes smaller, increasing the likelihood of stochastic extinction.
This ‘niche-width / area tradeoff” is analogous to the area-heterogeneity tradeoff in the sense that in finite areas, mechanisms that initially contribute to increased species richness (heterogeneity in Allouche et al. 2012, niche width here) are the very same drivers of local extinction, as long as patch isolation prevents rescue effects and patches are relatively small.”

It seems to me that these two mechanisms might be non-analogous. In line with this, maybe the suggested trade-off between niche width effects and area effects (“and the area available for populations in patches becomes smaller”) is a bit controversial or unclear. It seems to me that habitat areas remain the same when you simply modify the inter-patch migration parameter. Of course, increased inter-patch migration facilitates “invasion and inter-specific space competition”, increasing extinction of low abundance populations and reducing diversity, but this effect does not necessarily require a trade-off with area, although area effects and trade-offs might also occur.

238-262. This is just a minor suggestion. Please avoid excessive speculation in the last part of the discussion. I found that the content of the last two paragraphs is often out of the scope of the manuscript and perhaps is a bit too speculative.

Stein, A., Gerstner, K., & Kreft, H. (2014). Environmental heterogeneity as a universal driver of species richness across taxa, biomes and spatial scales. Ecology letters, 17(7), 866-880.

Carnicer, J., Brotons, L., Herrando, S., & Sol, D. (2013). Improved empirical tests of area-heterogeneity tradeoffs. Proceedings of the National Academy of Sciences, 110(31), E2858-E2860

Carnicer, J., Brotons, L., Sol, D., & De Cáceres, M. (2008). Random sampling, abundance–extinction dynamics and niche‐filtering immigration constraints explain the generation of species richness gradients. Global Ecology and Biogeography, 17(3), 352-362.

Additional references:

Carnicer, J., Brotons, L., Stefanescu, C., & Penuelas, J. (2012). Biogeography of species richness gradients: linking adaptive traits, demography and diversification. Biological Reviews, 87(2), 457-479..

Carnicer, J., Brotons, L., Sol, D., & Jordano, P. (2007). Community‐based processes behind species richness gradients: contrasting abundance–extinction dynamics and sampling effects in areas of low and high productivity. Global Ecology and Biogeography, 16(6), 709-719.

---

## Round 0.2 · Minor Revisions

Reviewer 1 has some specific suggestions that you should seriously consider. From my own reading, I found the model description improved, but the math writing can be improved; in particular you have equation [1] which cannot be a valid equation for whole numbers. So under details I suggest a modification.

Details

L 100 can host -> hosts (as all patches are occupied, which is essential for the logic of eq [1], in particular the first term with square brackets).
L 105 delete “initially” as the number of individuals per patch does not change.

L109-110. This becomes the first term on the rhs of [1]. But it should be a whole number, so how was this done.

L113-128. Equation [1] cannot be a valid equation for whole numbers. I suggest the following rewrite:
Merge line 113/114 with line 127/128 and next describe the terms needed in the multinomial distribution.

·

Basic reporting

no comments

Experimental design

no comments

Validity of the findings

Thanks for testing the difference in outcome after 1000 time steps instead of 250. As you say, "no evidence for the emergence of unimodal relationships due to different within-patch dynamics" (slow or fast). However, the comparison of species richness values (fig. S2) does indicate that especially for larger sigma and z values the species richness changed considerable between t=250 and t=1000. Thus, it does not seem entirely valid to consider the results as related to equilibrium state. I suppose the current figure 2 is still based on 250 time steps. It would be interesting to add a similar figure for t=1000 to the suppl. info. as I think the time aspect can not completely ignored.
As a side-note, it is not so straightforward to interpret this figure S2: do the settings with little change between 250 and 1000 imply that equilibrium was quickly reached already, or is equilibrium far away but approached very slowly? I guess only time series of species richness can explain what is happening...

Additional comments

- some of the equations seem a bit mangled (by the pdf conversion?)
- 112-122 is one long sentence. Maybe split it into several, for readability?
- I tried running the code in R. Managed to do so only after out-commenting the statement " Jpatches = reorder(Jpatches)" but I don't know whether the model is still correct after doing so....

---

## Round 0.3 · Minor Revisions

Thank you for taking care of the details in model description and the software code. I have one more small point. On line 144, you draw from a multinomial distribution for which you specify the probabilties but not the size (number of draws). I suggest you add after "each species", "and as sample size the total number of died individuals in the patch." And I would prefer the asterisk instead of the inverted question mark above the N's. After these changes the manuscript is accepted.

---

## Round 0.4 · accepted · Accept

the paper is now in fine shape and will generate, I hope, some discussion! Thanks for putting attention to the details.